# A Community-Based Intervention in Middle Schools in Spain to Improve HPV Vaccination Acceptance: A “Pill of Knowledge” Approach

**DOI:** 10.3390/vaccines14010022

**Published:** 2025-12-24

**Authors:** Ernesto J. González-Veiga, Sergio González-Palanca, Gerardo Palmeiro-Fernández, Juan C. Domínguez-Salgado, Paula Rubio-Cid, María López-Pais, Vito Carlo Alberto Caponio, Ellen M. Daley, Alejandro I. Lorenzo-Pouso

**Affiliations:** 1Primary Health Care Unit, EOXI Santiago de Compostela and Barbanza, Galician Health Service, 15706 A Coruña, Spain; ernestojgv@gmail.com; 2Department of Gynecology & Obstetrics, Valdeorras Hospital, El Barco de Valdeorras, 32300 Ourense, Spain; sergiojosegp@gmail.com (S.G.-P.); juan.carlos.dominguez.salgado@sergas.es (J.C.D.-S.); 3Primary Health Care Unit, EOXI Ourense, Verín and El Barco de Valdeorras, Galician Health Service, 32300 Ourense, Spain; gerardo.palmeiro.fernadez@sergas.es; 4Department of Gynecology & Obstetrics, Álvaro Cunqueiro Hospital, 36312 Vigo, Spain; paula.rubio.cid@sergas.es; 5Department of Gynecology & Obstetrics, University Clinical Hospital of Santiago de Compostela, 15706 A Coruña, Spain; maria.lopez.pais@sergas.es; 6Department of Life Sciences, Health and Health Professions, Link Campus University, Via del Casale Di San Pio V 44, 00165 Rome, Italy; 7College of Public Health, University of South Florida, Tampa, FL 33612, USA; edaley@usf.edu; 8Oral Medicine, Oral Surgery and Implantology Unit, Faculty of Medicine and Odontology, University of Santiago de Compostela, 15782 Santiago de Compostela, Spain; alexlopo@hotmail.com; 9ORALRES Group, Health Research Institute of Santiago de Compostela (IDIS), Universidade de Santiago de Compostela, 15706 Santiago de Compostela, Spain

**Keywords:** behavioural health, human papillomavirus, vaccination, decision-making, health literacy, public health

## Abstract

**Objectives:** Despite high overall vaccination coverage in Galicia, Spain, human papillomavirus (HPV) vaccine uptake remains below the 90% target set by the World Health Organization for 2030. This study aimed to assess baseline knowledge of HPV and attitudes towards HPV vaccination among Galician adolescents and to evaluate the impact of a brief educational intervention delivered as a “pill of knowledge”. **Methods:** A quasi-experimental pre-/post-intervention study was conducted among 967 students aged 12–16 years from 16 secondary schools in Galicia during the 2023–2024 academic year. A concise, structured 15-min educational session termed a “pill of knowledge” was delivered, and HPV-related knowledge and vaccination intention were measured immediately before and after the intervention using a standardized questionnaire. **Results:** Following the “pill of knowledge”, the mean proportion of correct responses increased by 30.1 ± 16.6% across all knowledge items. Among unvaccinated participants, intention to accept HPV vaccination rose from 77.7% to 94.4% in girls and from 64.7% to 85.8% in boys. Pre-intervention predictors of vaccination intention included perceived vaccine efficacy and baseline HPV knowledge. Post-intervention independent predictors comprised being female, younger age (12–13 years), and prior sexual education delivered by teachers or parents. The overall predictive accuracy of the logistic regression model for vaccination intention improved from 75.6% before the intervention to 92.7% afterwards. **Conclusions:** A brief, school-based “pill of knowledge” produced substantial and immediate improvements in HPV knowledge and vaccination acceptance among Galician adolescents. These findings strongly support the systematic incorporation of short, evidence-based educational interventions of this kind into the school setting as an effective public health measure to increase HPV vaccine coverage and advance progress toward WHO elimination targets.

## 1. Introduction

The human papillomavirus (HPV) is widely recognized as a major aetiological agent in the development of malignant diseases, including cervical, anogenital, and head and neck cancers [1]. HPV infection constitutes the most prevalent sexually transmitted infection worldwide, with approximately 80% of sexually active individuals acquiring it at some point during their lifetime. Although most infections (around 90%) are asymptomatic and cleared spontaneously within two years, persistent infection in a minority of cases can lead to benign lesions or progress to malignancy in anogenital and oropharyngeal sites [2].

The World Health Organization (WHO) has established the ambitious target that 90% of girls should be fully vaccinated against HPV by the age of 15 years by 2030 as part of its global strategy to mitigate cervical cancer [3]. In Galicia, HPV vaccination was incorporated into the regional immunization schedule in 2008, initially targeting all girls born from 1994 onwards. Although female vaccination coverage in Galicia remains among the highest in Spain (78.5% in 2023, with a 13-year mean of 79.8%), the WHO-recommended threshold has not yet been achieved, a pattern observed across numerous European countries. To address residual gaps, the Galician Health Service (SERGAS) has extended free vaccination to all women born since 1994 who were previously unvaccinated. Since September 2022, vaccination has also been offered to boys born from 2010 onwards, although uptake data for this cohort are not yet available.

During the study period, HPV vaccination in Galicia was provided free of charge, included in the official schedule, and accessed either through self-appointment via the SERGAS mobile application or by requesting an appointment at the corresponding primary care center. Public Health authorities instructed primary care teams to opportunistically promote vaccination to adolescents (and their families) during any clinical encounter [4].

Evidence from non-interventional studies consistently indicates that knowledge of HPV and vaccine acceptability among adolescents remains modest, underscoring the pivotal role of accurate information in improving coverage [5,6]. Many adolescents fail to identify HPV as a sexually transmitted infection, are unaware of its association with genital warts or cervical cancer, and do not recognise that both sexes are affected [7,8]. Improvements in HPV-related knowledge have been demonstrated following structured educational interventions in community settings [9,10]. However, no comparable interventional studies from Spain were identified in the reviewed literature. Moreover, data are scarce regarding the effect of brief educational interventions on vaccination intention among adolescents of both sexes, particularly within the Spanish context [11,12].

Accordingly, the present study was designed to: (i) assess baseline HPV knowledge and attitudes towards vaccination among adolescents in our region, (ii) develop a targeted brief educational intervention termed a “pill of knowledge”, and (iii) evaluate its immediate impact on participants’ knowledge, attitudes, and intention to accept HPV vaccination.

## 2. Materials and Methods

### 2.1. Study Design

A quasi-experimental pre-post intervention study was conducted amongst a representative sample of male and female adolescents attending secondary schools in Galicia, an autonomous region in northwestern Spain with independent Health and Education systems, between October 2023 and February 2024. The study protocol received ethical approval from the Pontevedra-Vigo-Ourense Research Ethics Committee (Registration Code 2023/215) and was authorised by the Galician Ministry of Culture. This study adhered to STROBE guidelines [13].

### 2.2. Study Population and Recruitment

Sixteen secondary schools were selected, with four from each of the region’s four provinces, representing both coastal and inland areas; nine were urban and seven semi-urban or rural settings. The study enrolled students of all genders aged 12–16 years. This age range was chosen as it corresponds to pupils enrolled in secondary education. Eligible students were required to be proficient in Spanish and Galician, and capable of reading, comprehending, and responding to the questionnaire independently. The principal investigator approached school directors selected through convenience sampling based on investigator and teaching staff availability. The study objectives and methodology were explained in accordance with the protocol approved by the Ethics Committee.

Class tutors informed all secondary school students that a medical professional (investigator) would deliver an educational presentation on HPV at a scheduled date, with pre- and post-intervention assessments to evaluate knowledge acquisition. All eligible students were invited to participate, and informed consent documentation was provided. The teacher responsible for each student group recommended that participants verify their HPV vaccination status, ensuring receipt of the appropriate number of doses: two doses for those aged < 15 years or three doses for those aged ≥ 15 years. This information was also emphasized in the consent documentation provided. In accordance with Ethics Committee stipulations, both vaccinated and unvaccinated adolescents were eligible for inclusion; vaccination status was not verified through medical records, and questionnaires were administered anonymously. Written informed consent was obtained from all participants and their parents or legal guardians who voluntarily agreed to participate. The school director designated a member of teaching staff to serve as liaison and coordinator for the research activities with the investigators. Participants were classified as “losses to follow-up” if they declined participation after invitation, lacked parental consent, failed to complete the pre-intervention questionnaire, did not attend the educational session, did not respond to the post-intervention assessment, fell outside the specified age range, or provided multiple responses to single-answer questions.

### 2.3. Sample Size Calculation

The study by Liu et al. [10] provided essential data for sample size estimation. Their study included 1675 students aged 10–14 years, of whom 55.2% expressed willingness to receive vaccination. Due to the absence of local data on vaccination intention, we conducted a pilot study to estimate vaccination willingness and evaluate the comprehension and reliability of the pre- and post-intervention assessments. Whilst Liu’s study was not conducted in the same geographical region, its methodology and design were sufficiently similar to permit comparison with our investigation.

Two additional schools were randomly selected for the pilot phase, one urban and one semi-rural, yielding 131 participants (78 females, 52 males, and one non-binary individual). Vaccination willingness was calculated overall and stratified by gender and age. The observed proportion of 61.7% was comparable to that reported by Liu et al. (*p* = 0.39) [10]. Using Epidat© 4.2 (https://www.sergas.es/Saude-publica/EPIDAT-4-2, accessed on 9 September 2023) and Granmo© 8.0 (https://www.datarus.eu/aplicaciones/granmo/, accessed on 9 September 2023) calculators for sample size determination in paired proportion comparisons, minimum sample sizes were calculated by age and gender to detect a vaccination intention of 95%, with a minimum statistical power of 80% and significance level of 0.05. Assuming a 10% loss to follow-up rate, minimum paired sample sizes ranged from 24 in the male adolescent group to 66 in the female pre-adolescent group. For the third group of non-binary participants, the required number of pairs did not exceed six in any stratum. Cronbach’s alpha coefficient for internal consistency of the pilot instrument was 0.78 (95% CI [0.71, 0.83]).

The final study enrolled a non-probability convenience sample of 967 participants, stratified by gender and age, representing a participation rate of 82.9%.

### 2.4. Variables Collected and Intervention Protocol

A survey instrument was developed comprising eight variables on demographic characteristics (age, gender, residence) and subjective levels of sexual health information and prior HPV knowledge, primary source of sexual health information, prior complete HPV vaccination status, parental educational attainment, plus 17 identical pre- and post-intervention variables concerning HPV infection epidemiology, symptomatology, diagnosis, treatment and prevention, as well as vaccine safety, efficacy, delivery setting, and vaccination willingness (see Appendix A for complete questionnaire). These items were derived from previous instruments developed by our research group and others [6,8].

As previously described, questionnaire content validity was established through a modified two-round Delphi process conducted in accordance with published methodological guidelines for Delphi studies [14] and following the same procedure successfully employed by our research group in prior vaccine-related studies [6]. An expert panel was formed consisting of six experienced general practitioners and epidemiological researchers. In the first round, each panellist independently evaluated the relevance, clarity, and appropriateness of every item using a 5-point Likert scale (1 = strongly disagree; 5 = strongly agree). The median score across all items was 4.86 (interquartile range 4–5), with a mean of 4.86 ± 0.31. No panellist suggested modifications or additional items, and all participants explicitly indicated that a second round was unnecessary. Consensus was therefore achieved after the first round, and the questionnaire was considered fully validated for use in the study.

Pre-intervention: At the educational centre, in a large, adapted room, following a brief explanation, the teacher distributed sealed envelopes containing two additional envelopes to students. Participants opened envelope 1 and completed the pre-intervention assessment over 15 min in the absence of a healthcare professional to minimize potential authority bias and Hawthorne effect. Subsequently, the teacher collected the complete questionnaires [15].

Intervention: The investigator delivered a 15-min PowerPoint presentation containing evidence-based information about HPV (a “knowledge capsule”). Content included HPV definition, epidemiology, transmission routes, prevention strategies, associated diseases, diagnosis, vaccination schedule, vaccine efficacy and safety, and potential adverse effects. During the intervention, the investigator clarified that females born after 1994 who were unvaccinated or had not received both doses could access the vaccine free of charge if desired. Similarly, males born in 2010 or later could access free vaccination should they wish. Following the presentation, students were given a 15-min break to minimize immediate recall bias [16].

Post-intervention: Students completed the post-intervention assessment (envelope 2), which contained identical items to the pre-intervention questionnaire and was coded with the same identification number, again in the absence of the healthcare professional. After 15 min of completion, the teacher collected the questionnaires. As a complementary educational activity, following questionnaire collection, group discussions were conducted at all schools with the investigator present.

### 2.5. Statistical Analysis

Quantitative variables were expressed as means and standard deviations (±SD), and qualitative variables as frequencies and percentages (N, %); 95% confidence intervals were calculated throughout. Student scores were averaged to facilitate presentation, and mean change scores for multi-item scales were obtained by subtracting pre- from post-intervention means. Normality was assessed with the Shapiro–Wilk test, and parametric or non-parametric tests were applied accordingly. Students were stratified by age (12–13 and 14–16 years) and gender following Zimmerman et al. [12]. Logistic regression models were developed to identify predictors of vaccination in pre- and post-intervention phases, and ANCOVA models were used to compare mean values of significant predictors while controlling for confounders. Multicollinearity was assessed using variance inflation factors and tolerance statistics, with no evidence of problematic collinearity. All analyses were performed using IBM SPSS Statistics 27.0.1.0 (IBM Corporation, Armonk, NY, USA), with the statistician blinded to study hypotheses and α set at 0.05.

## 3. Results

### 3.1. Participants

A total of 1167 invitations and consent forms were distributed across the participating centres. Of these, 967 students met the inclusion criteria (82.9%) (Figure 1).

A total of 497 participants (51.4%) were identified as girls [184 (37.0%) aged 12–13 years and 313 (63.0%) aged 14–16 years], and 452 participants (46.7%) were identified as boys [155 (34.3%) aged 12–13 years and 297 (65.7%) aged 14–16 years]. In addition, 18 participants (1.9%) identified as non-binary [11 (61.1%) aged 12–13 years and 7 (38.9%) aged 14–16 years]. Basic descriptive data are summarised in Table 1.

### 3.2. Sources of Sexual Information and Perceived Sexual Knowledge

The most common sources of sexual information for both girls and boys were parents (30.9% of girls, n = 497; 28.1% of boys, n = 452; *p* > 0.05) and teachers (20.5% of girls; 23.9% of boys; *p* > 0.05). A similar pattern was observed in both preadolescents and adolescents, although among those aged 14–16 years, social media played a more prominent role than in the 12–13-year-old group (*p* < 0.001) (Figure 2). Parents with secondary or higher education were more likely to provide sexual information to their children (67.7%, n = 282, compared with 2.4% among those with only primary education; *p* < 0.001). No significant differences were observed between boys and girls regarding the amount of sexual information perceived to have been received. However, adolescents aged 14–16 years reported feeling more informed than those aged 12–13 years (*p* < 0.001). Urban residents (*p* = 0.002) and those receiving information from parents, teachers, or social media (*p* < 0.001) also reported higher levels of perceived knowledge.

### 3.3. Perceived HPV Knowledge and HPV Vaccination

All genders reported having heard of HPV at similar rates, although this was more common in the 14–16-year-old group (n = 967; 51.9% vs. 28.5% in the 12–13-year-old group; *p* < 0.001). Vaccination coverage was higher among girls than boys (71.0%, n = 497 vs. 36.0%, n = 452; *p* < 0.001). Vaccination status was not associated with the amount of sexual information received. A positive association was observed between being vaccinated and having heard of HPV (593.5; *p* < 0.001).

### 3.4. Perceived HPV Vaccine Efficacy and Safety

Perceived vaccine efficacy increased in the post-intervention phase (r = 0.3; *p* < 0.001) (Figure 3). Initially, perceived efficacy did not differ between the two age groups; however, following the intervention, it was significantly higher in the 14–16-year-old group (*p* = 0.04). Higher perceived efficacy was also observed among participants who had previously heard of HPV (r = 0.2; *p* < 0.001). At baseline, vaccinated participants attributed greater efficacy to the vaccine than those who were unvaccinated (*p* < 0.001), although this difference was no longer present after the intervention.

Initially, 357 girls (71.8%) perceived the HPV vaccine as safe compared with 272 boys (60.1%) (*p* < 0.001). This perception increased significantly in both genders after the intervention, reaching 96.1% among girls and 87.8% among boys (n = 478), with the gender difference reduced but still significant (*p* < 0.001) (Figure 4). Regarding age, adolescents aged 14–16 years (n = 423, 68.6%) considered the vaccine to be safer than those aged 12–13 years (n = 213, 60.1%; *p* = 0.02), although this difference disappeared following the intervention. At baseline, perceived safety was also higher among participants who had previously heard of HPV (*p* < 0.001). In both phases, vaccinated participants (n = 524) attributed greater safety to the vaccine—before the intervention (73.3% vs. 56.8% among unvaccinated participants; *p* < 0.001) and after the intervention (95.0% vs. 89.4%; *p* < 0.001). Higher perceived safety was positively associated with higher perceived efficacy in both phases (*p* < 0.001).

### 3.5. Questionnaire Performance

The overall mean percentage of correct responses was higher in the post-intervention phase (95% CI [28.8, 31.3]; *p* < 0.001). Girls achieved more correct answers both before and after the intervention (95% CI [3.8, 9.0]; *p* < 0.001 and 95% CI [3.7, 7.6]; *p* < 0.001, respectively). Before the intervention, the 14–16-year-old group obtained more correct answers than the 12–13-year-old group (95% CI [3.8, 9.2]; *p* < 0.001), although no differences were observed between the two age groups in the post-intervention phase. Improvement was significantly greater in the 12–13-year-old group than in the 14–16-year-old group (95% CI [2.6, 7.8]; *p* < 0.001). Participants who had previously heard of HPV obtained a higher number of correct responses (r = 0.2; *p* < 0.001). Vaccinated participants achieved a higher frequency of correct answers in both phases (95% CI [1.0, 1.9]; *p* < 0.001 and 95% CI [0.5, 1.2]; *p* < 0.001), although the unvaccinated group showed a greater improvement after the intervention (Table 2).

A higher frequency of “I don’t know” responses was reported by boys than by girls in both the pre-intervention phase (95% CI [2.8, 8.1]; *p* < 0.001) and the post-intervention phase (95% CI [2.6, 5.5]; *p* < 0.001). The overall percentage of “I don’t know” responses decreased from 25.4% before the intervention to 3.5% afterwards, with no gender differences observed in the post-intervention phase. Initially, the 12–13-year-old group selected “I don’t know” more often than the 14–16-year-old group (95% CI [3.5, 9.7]; *p* < 0.001), although no age-related differences were identified following the intervention.

Results for all items in both phases are shown in Figure 5. For every item, a significantly higher number of correct responses was observed in the post-intervention phase than in the pre-intervention phase (95% CI [28.8, 31.3]; *p* < 0.001).

### 3.6. Vaccination Intentions

Before the intervention, 773 participants (79.9%) had been vaccinated or intended to be vaccinated. After the intervention, 897 participants (92.8%) accepted vaccination (*p* < 0.001). This intention increased across all gender and age strata (Table 3).

Thirteen participants (1.3%) changed from acceptance to refusal, with no association observed with perceived vaccine efficacy or safety; they showed no improvement in knowledge and obtained significantly lower scores than those willing to be vaccinated (*p* < 0.001).

Overall, 57 adolescents (5.8%; 42 boys and 10 girls) remained undecided or opposed, 80% of whom were older. Compared with participants willing to be vaccinated, this group demonstrated lower HPV knowledge (*p* < 0.001) and perceived lower vaccine safety at baseline (17.6% vs. 3.6%; *p* < 0.001). Although knowledge improved post-intervention, their scores remained significantly lower in both phases (*p* < 0.001).

Of the 967 participants, 443 (45.8%) were unvaccinated or uncertain of their vaccination status (144 girls, 289 boys, and 10 non-binary participants). Girls showed a higher intention to be vaccinated than boys both before the intervention (77.7% vs. 64.7%; *p* = 0.008) and after (94.4% vs. 85.8%; *p* = 0.01), with significant increases observed among genders (*p* < 0.001) (Figure 6). Intention also increased in both age groups (12–13 years: 94.6% vs. 80.6%; 14–16 years: 91.7% vs. 79.6%; *p* < 0.001 for both).

A greater proportion of boys than girls remained hesitant post-intervention (14.2% vs. 5.5%; *p* = 0.01), with six participants shifting from acceptance to refusal. Those refusing vaccination continued to achieve lower correct response scores in both phases (*p* < 0.001).

### 3.7. Logistic Regression Analysis of Predictors of Vaccination Intention

In the pre-intervention phase, vaccination intention was predicted by perceived vaccine efficacy and knowledge level, with higher likelihood observed among participants who answered 6–11 or ≥12 questions correctly. The model achieved an accuracy of 75.6%. Following the intervention, predictors included gender, age, and sexual education received from teachers and/or parents. This model explained 45.2% of the variance and improved prediction accuracy to 92.7%. Detailed results of these analyses are presented in Table 4.

## 4. Discussion

To our knowledge, this is the first study to implement a brief educational intervention to improve knowledge, attitudes, and practices regarding HPV among Spanish secondary school students. Most previous research has focused on adult women [17,18], parental attitudes [6,19,20], or girls older than 16 years [21,22]. Although vaccination decisions are generally made by parents, children’s opinions can influence uptake, as reported in China (73.6%) [23], and some parents believe informed adolescents should be able to request the vaccine [22]. In Europe, except Finland, decisions rest with parents [8], whereas Canadian data suggest involving both parents and adolescents [24]. These dynamics justify focusing on adolescents.

Knowledge of HPV did not differ by gender in our study, consistent with Wang [25]. Among 14–16-year-old girls, awareness levels were similar to those reported in Valencia for 15-year-olds (89.9%) [26], and higher than in Italy (70%) [8,9] or Hong Kong (44%) [27]. Awareness of the HPV vaccine was also higher than reported in Zhang’s meta-analysis (15.9%) [28], aligning with other publications [9,26]. Participants identified parents, teachers, and social media as primary sources of sexual health information, whereas other studies highlight social media, internet, television, and the press as predominant sources [25,29].

Vaccination coverage exceeded 60% in girls and nearly 50% in boys, comparable or superior to Sweden (58.8%) [11], China (33.5%) [25], and France (10.1%) [30], and consistent with other Spanish surveys (74.5% in girls) [26]. Most students knew the routes of HPV transmission, in line with previous research [10,26,31]. Many participants trusted vaccine efficacy, as reported in a Spanish survey where 67% of girls considered it highly effective against cancer [26], and lack of knowledge about efficacy and safety has been identified as a key barrier in meta-analytic evidence [28]. Baseline perceptions of vaccine safety were acceptable but increased significantly post-intervention, mirroring findings in Italy (85.2% perceived the vaccine as safe) [8] and Spain [26,32].

Knowledge improved across all questions following the intervention, supporting its effectiveness, consistent with other studies [11,33,34]. Unvaccinated students also achieved higher post-intervention scores, in line with prior reports [28,30,34]. Younger students (<13 years) initially scored lower [31], while girls generally scored higher across phases, consistent with Grandahl [11], and vaccinated girls outperformed unvaccinated peers [26]. Knowledge gains were associated with increased willingness to vaccinate, as described by Zhang [28], with adequate knowledge doubling vaccination rates in Brazil [34]. Higher knowledge correlated with parental education and social media use, whereas lower knowledge was associated with mothers with primary education [8].

In the present study, individual follow-up to assess knowledge retention was not possible. However, similar interventions have shown lasting effects: Liu’s study reported that knowledge and willingness to vaccinate remained higher one-year post-intervention compared with controls [10], and a Mexican study found that adolescents’ post-intervention knowledge stayed elevated after one year, with no significant decline from immediately post-intervention [31]. These findings suggest that brief educational interventions can have enduring impacts on knowledge and vaccination attitudes.

Among unvaccinated participants, intention to vaccinate increased significantly post-intervention, approaching WHO-2030 targets for girls, with comparable findings in other studies [9,10,23,28]. Decision-tree and regression analyses indicate that positive attitudes, knowledge, and age ≥ 12 years are strong predictors of vaccination intention [10,34].

Given the association between low education and higher HPV prevalence [35], it is critical to prioritise adolescent education. Healthcare professionals should provide accurate information on HPV and vaccination, as their knowledge surpasses that of parents and can correct myths [36,37], with sessions delivered by professionals appearing more effective than those led by educators [38,39]. Brief, 15-min “pill of knowledge” interventions are practical, low-cost, and effective at increasing vaccination intention, especially in school settings, which are ideal for sexual health education [10,40]. Based on our findings, we recommend periodic educational sessions delivered by healthcare professionals.

Key limitations of the study include limited external validity due to its geographic restriction to a single northern region of Spain and the use of a convenience sampling strategy for school selection, although no statistically significant differences in gender or age distributions were observed across participating schools. The inclusion of a control group was initially planned but ultimately not implemented following the Ethics Committee’s recommendation to ensure all students had access to the intervention, which may limit the ability to fully attribute observed effects to the educational programme. The low number of participants who declined to participate precluded formal analysis of refusal characteristics; nevertheless, no systematic differences in gender or age are anticipated, given their enrolment within the same school cohorts. Selection bias toward pro-vaccination families is considered unlikely, as the observed vaccination coverage in the sample was lower than national estimates. Potential information and observation biases, including recall bias, authority bias, and observer influence, were mitigated by instituting a brief washout period between pre- and post-assessments and by ensuring researcher absence during questionnaire administration. While the overall sample size was adequate to detect pre/post differences, a larger cohort would have enabled more robust exploration of additional covariates and subgroup analyses, particularly among participants who remained hesitant or unwilling to vaccinate following the intervention. Finally, a key shortcoming is the lack of evidence on the durability of the intervention’s effects. Although the “pill of knowledge” increased short-term knowledge and intention, we do not know whether these gains persist or translate into actual vaccination behaviour, and no regional post-intervention data were available to assess real-world impact.

In summary, this topic is timely and relevant for public health, particularly in the European context of improving vaccination coverage to meet WHO-2030 targets. A brief “pill of knowledge” intervention is simple, replicable, practical, and low-cost, making it a tool with potential for policy formulation and routine clinical practice.

## 5. Conclusions

Our results show that a brief school-based “pill of knowledge” intervention, providing accurate and evidence-based information, can enhance adolescents’ intention to receive the HPV vaccine and improve their understanding of HPV-related diseases, including prevention and diagnosis. Further research is warranted to evaluate and strengthen educational initiatives aimed at increasing HPV knowledge and awareness and, ultimately, improving vaccination uptake.

## Figures and Tables

**Figure 1 vaccines-14-00022-f001:**
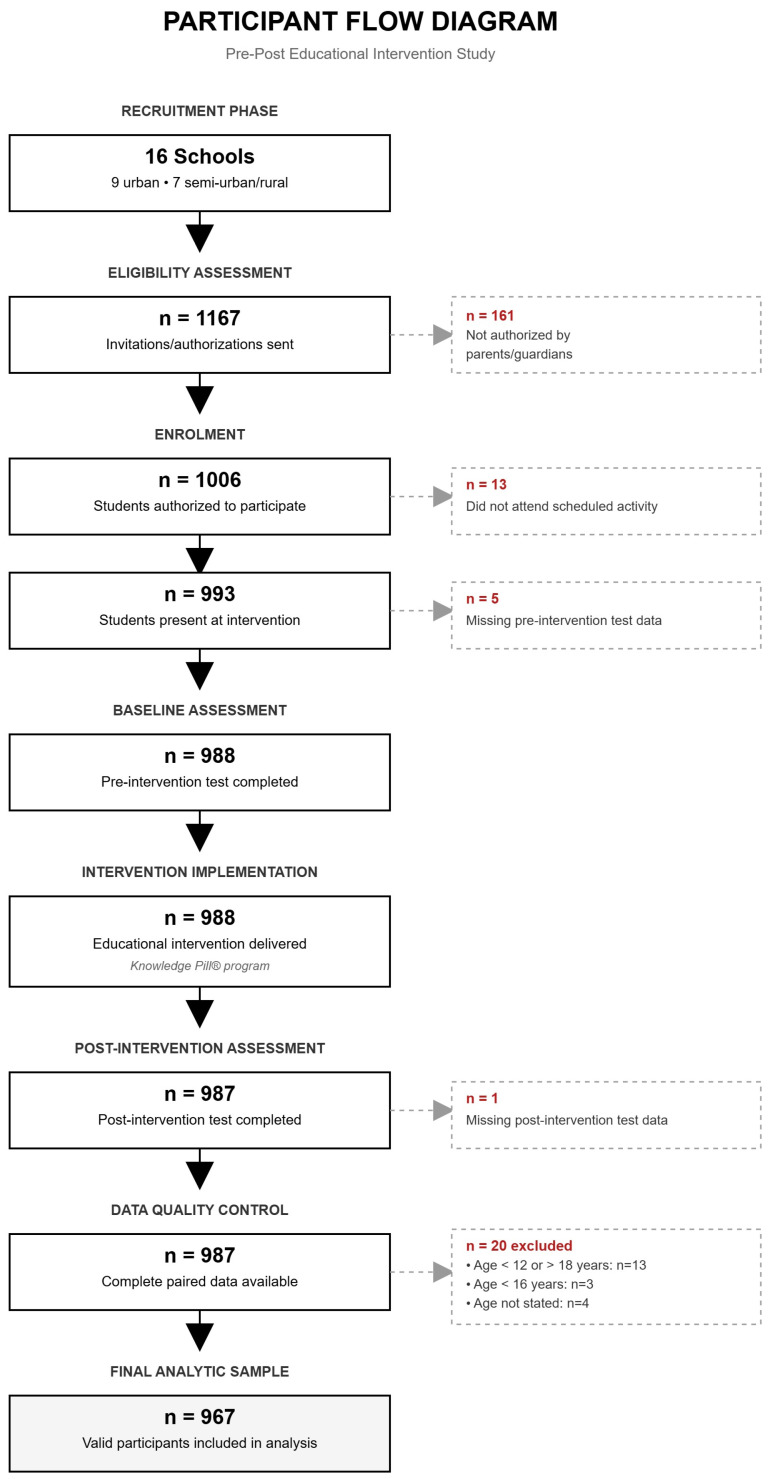
Flow diagram illustrating participant progression through all phases of the educational intervention study.

**Figure 2 vaccines-14-00022-f002:**
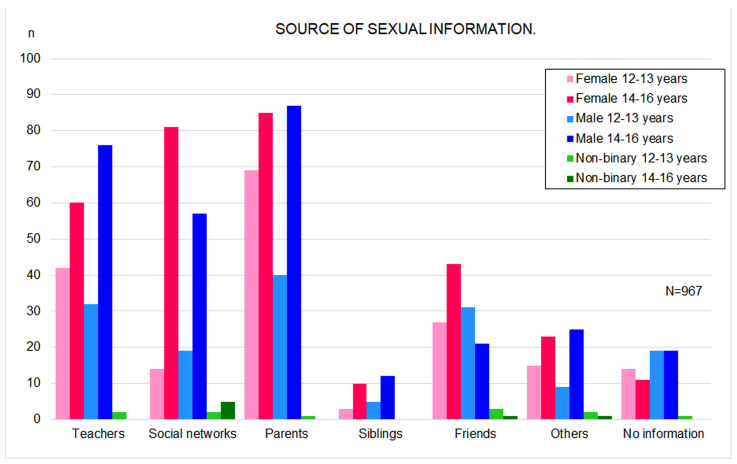
Bar Chart of Sources of Sexual Information by Gender and Age Group.

**Figure 3 vaccines-14-00022-f003:**
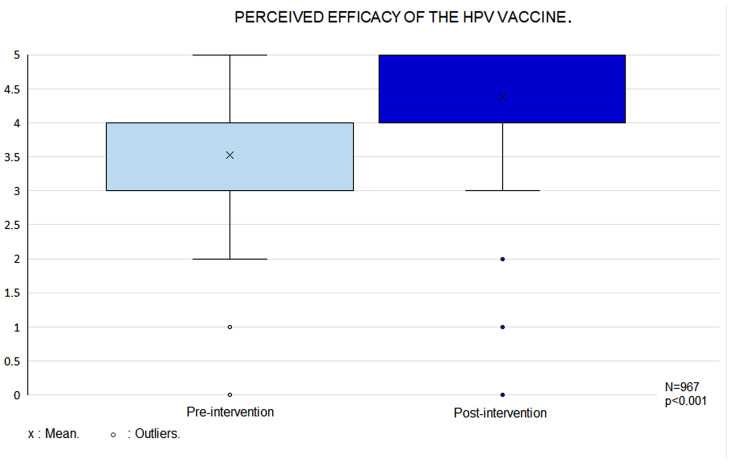
Box Plot of Perceived HPV Vaccine Efficacy Before and After the ‘Pill of Knowledge’ Intervention.

**Figure 4 vaccines-14-00022-f004:**
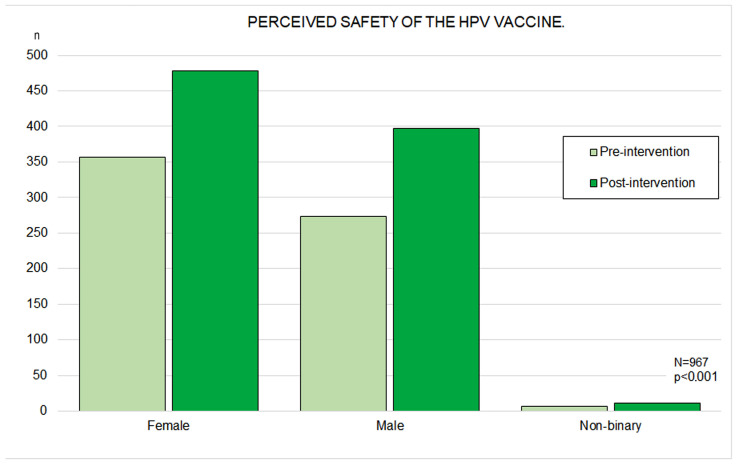
Bar Chart of Perceived HPV Vaccine Safety Before and After the ‘Pill of Knowledge’ Intervention, Stratified by Gender.

**Figure 5 vaccines-14-00022-f005:**
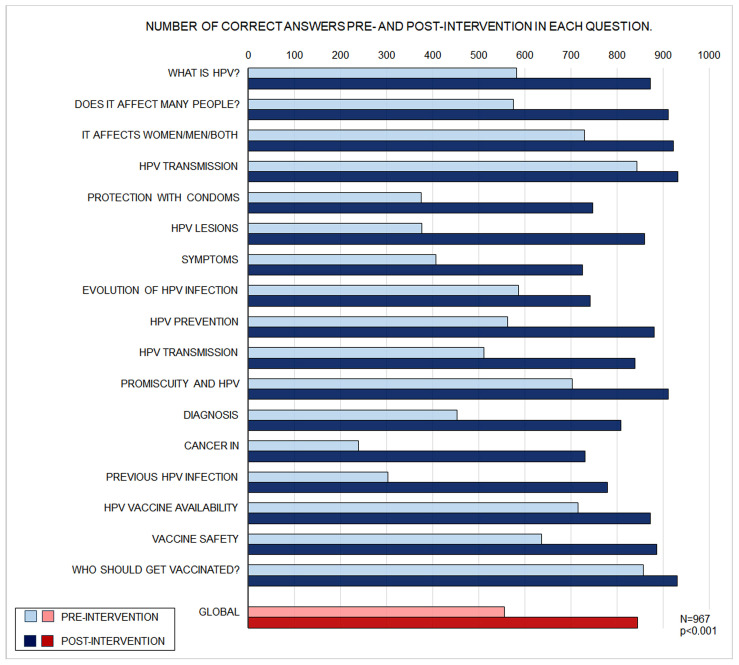
Bar Chart of Number of Correct Answers Pre- and Post-Intervention for Each Question.

**Figure 6 vaccines-14-00022-f006:**
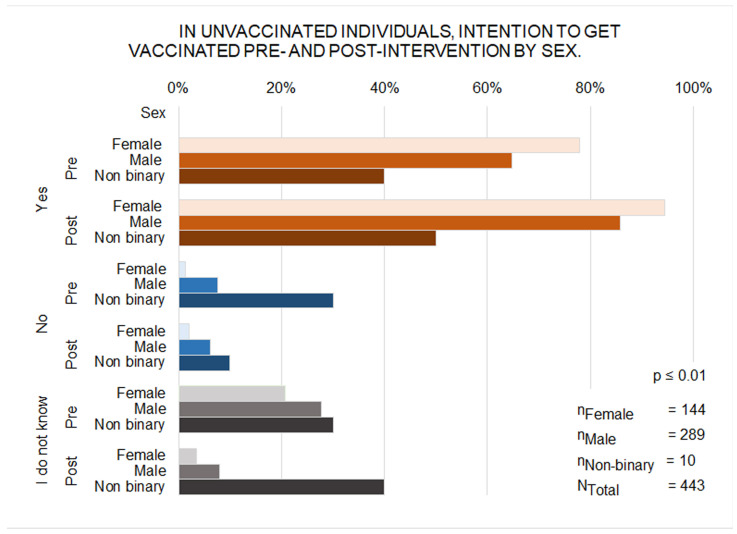
Bar Chart of Vaccination Intention in Unvaccinated Individuals Pre- and Post-Intervention by Gender.

**Table 1 vaccines-14-00022-t001:** Demographic Characteristics of Study Participants by Age Group and Gender.

Characteristic	Category	12–13 Years			14–16 Years		
		Female *n* (%)	Male *n* (%)	Non-Binary *n* (%)	Female *n* (%)	Male *n* (%)	Non-Binary *n* (%)
**Residence**	Rural	43 (23.4)	28 (18.1)	4 (36.4)	63 (20.1)	60 (20.2)	1 (14.3)
	Semi-urban	35 (19.0)	29 (18.7)	0 (0.0)	55 (17.6)	42 (14.1)	0 (0.0)
	Urban	106 (57.6)	98 (63.2)	7 (63.6)	195 (62.3)	195 (65.7)	6 (85.7)
**Self-reported sexual** **education**	None	11 (6.0)	8 (5.2)	0 (0.0)	2 (0.6)	14 (4.7)	1 (14.3)
	Little	65 (35.3)	57 (36.8)	6 (54.5)	59 (18.8)	44 (14.8)	0 (0.0)
	Enough	97 (52.7)	70 (45.2)	3 (27.3)	206 (65.8)	187 (63.0)	3 (42.9)
	A lot	11 (6.0)	20 (12.9)	2 (18.2)	46 (14.7)	52 (17.5)	3 (42.9)
**Prior Subjective knowledge about HPV ***	0	6 (3.3)	10 (6.5)	1 (9.1)	11 (3.5)	32 (10.8)	1 (14.3)
	1	30 (16.3)	24 (15.5)	3 (27.3)	40 (12.8)	31 (10.4)	0 (0.0)
	2	46 (25.0)	47 (30.3)	4 (36.4)	74 (23.6)	65 (21.9)	2 (28.6)
	3	73 (39.7)	45 (29.0)	1 (9.1)	98 (31.3)	88 (29.6)	1 (14.3)
	4	17 (9.2)	18 (11.6)	0 (0.0)	64 (20.4)	48 (16.2)	2 (28.6)
	5	12 (6.5)	11 (7.1)	2 (18.2)	26 (8.3)	33 (11.1)	1 (14.3)
**Parents’ Education**	Primary	3 (1.6)	6 (3.9)	0 (0.0)	12 (3.8)	21 (7.1)	0 (0.0)
	Secondary	38 (20.7)	33 (21.3)	2 (18.2)	111 (35.5)	65 (21.9)	1 (14.3)
	University	66 (35.9)	60 (38.7)	3 (27.3)	101 (32.3)	106 (35.7)	4 (57.1)
	Unknown	77 (41.8)	56 (36.1)	6 (54.5)	89 (28.4)	105 (35.4)	2 (28.6)
**HPV Vaccination**	Yes	116 (63.0)	74 (47.7)	6 (54.5)	237 (75.7)	89 (30.0)	2 (28.6)
	No	38 (20.7)	32 (20.6)	1 (9.1)	18 (5.8)	116 (39.1)	3 (42.9)
	Not reported	30 (16.3)	49 (31.6)	4 (36.4)	58 (18.5)	92 (31.0)	2 (28.6)

* Prior subjective knowledge about HPV is measured on a scale from 0 to 5, with higher values indicating greater perceived prior knowledge. Note: Percentages may not sum to 100% due to rounding.

**Table 2 vaccines-14-00022-t002:** Increase in hit rates by vaccination status, age and age group pre- and post-intervention.

Vaccination Status	Gender	n	Baseline Hit Rate (% ± SD)	Post-Intervention Hit Rate (% ± SD)	95% Confidence Interval	*p*-Value	Improvement in Hit Rate (%)
**Vaccinated**	**Female**	353	62.9 ± 18.0	91.4 ± 9.6	26.7 to 30.3	<0.001	28.5 ± 16.9
	**Male**	163	58.0 ± 19.1	86.4 ± 14.4	25.4 to 31.4	<0.001	28.4 ± 19.1
	**Non-binary**	8	50.8 ± 25.5	71.1 ± 34.9	2.3 to 38.4	0.032	20.3 ± 21.6
**Non-Vaccinated**	**Female**	144	53.6 ± 20.5	86.7 ± 12.2	29.9 to 36.2	<0.001	33.1 ± 19.2
	**Male**	289	51.5 ± 22.7	83.3 ± 20.9	29.2 to 34.3	<0.001	31.7 ± 22.0
	**Non-binary**	10	43.8 ± 28.0	73.1 ± 27.2	5.1 to 53.7	0.023	29.4 ± 34.0
**Vaccination** **Status**	**Age Group**	**n**	**Baseline Hit Rate (% ± SD)**	**Post-Intervention Hit Rate (% ± SD)**	**95% Confidence** **Interval**	** *p* ** **-Value**	**Improvement in Hit Rate (%)**
**Vaccinated**	**12–13**	196	56.3 ± 18.6	88.2 ± 12.4	29.3 to 34.6	<0.001	32.0 ± 18.9 ^a^
	**14–16**	328	64.2 ± 18.0	90.4 ± 12.4	24.4 to 27.9	<0.001	26.2 ± 16.5 ^a^
	**Total**	524	61.2 ± 18.6	89.6 ± 12.4	26.8 to 29.8	<0.001	28.3 ± 17.7 ^c^
**Non vaccinated**	**12–13**	154	48.6 ± 20.9	83.8 ± 16.1	24.4 to 27.9	<0.001	35.2 ± 22.3 ^b^
	**14–16**	289	53.9 ± 22.6	84.3 ± 0.1	28.1 to 32.9	<0.001	30.5 ± 20.8 ^b^
	**Total**	443	52.0 ± 22.1	84.1 ± 18.8	30.1 to 34.1	<0.001	32.1 ± 21.4 ^c^
**Global Total**		967	57.1 ± 20.8	87.1 ± 15.9	28.8 to 31.3	<0.001	30.1 ± 19.5

^a^ statistically significant difference (95% CI [2.7, 8.9]; *p* < 0.001) ^b^ statistically significant difference (95% CI [5.3, 8.9]; *p* < 0.03) ^c^ statistically significant difference (95% CI [1.3, 6.2]; *p* = 0.003).

**Table 3 vaccines-14-00022-t003:** Intention to Vaccinate by Vaccination Status, Gender, and Age Group.

Vaccination Status	Gender	Questionnaire Moment	Willingness to Vaccinate			Total
			Yes	No	Uncertain	
**Vaccinated**	**Female**	Pre-intervention	324 (91.7%)	3 (0.8%)	26 (7.4%)	353
		Post-intervention	348 (98.6%)	0 (0.0%)	5 (1.4%)	353
	**Male**	Pre-intervention	139 (85.3%)	4 (2.5%)	20 (12.3%)	163
		Post-intervention	155 (95.1%)	3 (1.8%)	5 (3.1%)	163
	**Non-binary**	Pre-intervention	7 (22.5%)	4 (12.9%)	20 (64.5%)	31
		Post-intervention	5 (38.5%)	3 (23.1%)	5 (38.5%)	13
**Non-Vaccinated**	**Female**	Pre-intervention	112 (77.8%)	2 (1.4%)	30 (20.8%)	144
		Post-intervention	136 (94.4%)	3 (2.1%)	5 (3.5%)	144
	**Male**	Pre-intervention	187 (64.7%)	22 (7.6%)	80 (27.7%)	289
		Post-intervention	248 (85.8%)	18 (6.2%)	23 (8.0%)	289
	**Non-binary**	Pre-intervention	4 (40.0%)	3 (30.0%)	3 (30.0%)	10
		Post-intervention	5 (50.0%)	1 (10.0%)	4 (40.0%)	10
**Vaccination Status**	**Age Group**	**Phase**	**Willingness to Vaccinate**			**Total**
			**Yes**	**No**	**Uncertain**	
**Vaccinated**	12–13 years	Pre-intervention	172 (87.8%)	2 (1.0%)	22 (11.2%)	196
		Post-intervention	189 (96.4%)	0 (0.0%)	7 (3.6%)	196
	14–16 years	Pre-intervention	298 (90.9%)	5 (1.5%)	25 (7.6%)	328
		Post-intervention	319 (97.3%)	4 (1.2%)	5 (1.5%)	328
**Non vaccinated**	12–13 years	Pre-intervention	110 (71.4%) ^a^	5 (3.2%)	39 (25.3%)	154
		Post-intervention	142 (92.2%) ^a,c^	5 (3.2%)	7 (4.5%)	154
	14–16 years	Pre-intervention	193 (66.8%) ^b^	22 (7.6%)	74 (25.6%)	289
		Post-intervention	247 (85.5%) ^b,c^	17 (5.9%)	25 (8.7%)	289
**Global Total**	967	Pre-intervention	773 (79.9%) ^d^	34 (3.5%)	160 (16.5%)	967
		Post-intervention	897 (92.8%) ^d^	26 (2.7%)	44 (4.6%)	967

^a^ statistically significant difference (95%CI [13.5, 28.1]; *p* = 0.034). ^b^ statistically significant difference (95%CI [13.4, 24.0]; *p* = 0.025). ^c^ statistically significant difference (95%CI [−13.1, −0.4]; *p* = 0.04). ^d^ statistically significant difference (95%CI [10.5, 15.2]; *p* < 0.001).

**Table 4 vaccines-14-00022-t004:** Predictors of Intention to Vaccinate on the basis of logistic regression analysis.

Phase	*Variable*	*B*	*SE*	*Wald*	*df*	*p*	*Exp(B)*	*95% CI for Exp(B)*
**Pre-Intervention**	**Highly effective vaccine**	0.972	0.464	4.388	1	0.036	2.644	[1.065, 6.567]
	**6–11 hits**	1.461	0.315	21.528	1	<0.001	4.309	[2.325, 7.985]
	**≥12 hits**	1.479	0.405	13.362	1	<0.001	4.389	[1.986, 9.702]
	**Constant**	−2.211	1.094	4.086	1	0.043	0.110	
	Hosmer–Lemeshow test: χ^2^= 4.218, df 8, *p* = 0.837; R^2^ = 0.279. Specificity: 92.1%; Sensibility: 40.0%; Forecast Accuracy: 75.6%							
**Post-Intervention**	**Female gender**	3.123	1.034	9.121	1	0.003	22.724	[2.993, 172.510]
	**Male gender**	2.158	0.951	5.144	1	0.023	8.653	[1.341, 55.857]
	**14–16 years old**	−1.189	0.544	4.775	1	0.029	0.304	[0.105, 0.885]
	**Sexual information: teachers**	1.678	0.758	4.898	1	0.027	5.356	[1.212, 23.676]
	**Sexual information: parents**	1.782	0.765	5.435	1	0.020	5.944	[1.328, 26.599]
	**Constant**	−2.908	1.520	3.659	1	0.056	0.055	
	Hosmer–Lemeshow test: χ^2^= 7.535, df 8, *p* = 0.480; R^2^ = 0.452. Specificity: 98.6%; Sensibility: 44.4%; Forecast Accuracy: 92.7%							

## Data Availability

The dataset supporting the conclusions of this article is available upon reasonable request by contacting the corresponding author.

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
