# Peer review of "A Community-Based Intervention in Middle Schools in Spain to Improve HPV Vaccination Acceptance: A “Pill of Knowledge” Approach"

_vaccines, 2025, doi:10.3390/vaccines14010022_

Round 1
Reviewer 1 Report
Comments and Suggestions for Authors
Dear authors, your manuscript addresses an important topic and is supported by current references. The research questions are clearly formulated. However, there are some improvements that could enhance the quality of presentation.
Introduction
The introduction is clear and concise. It describes the knowledge gap and rationale for the study. References are relevant and up to date.
Materials and Methods
This section presents important details, but some points need clarification:
Although the manuscript mentions that methodology details are provided in supplementary materials, the supplementary material appears to only contain the questionnaire. Meanwhile, the methods details are presented in the appendix included in the main text. Please clarify where additional methodological details can be found.
A flowchart detailing the study design, including participant selection and losses, would enhance understanding of the methodology.
The addition of a completed STROBE checklist is strongly recommended, given the observational study design (cited in the manuscript).
Results
The results section is comprehensive but may benefit from improved readability:
A participant flow diagram showing the number of students invited, included and excluded (losses) would better demonstrate the study flow.
The “subsections” within the tables appear visually complex and may be difficult to follow. Consider using other visual strategies.
The section feels dense in places. Could authors enhance readability?
There are no figures in the supplementary materials; you must include the referred figures.
Table legends appear to be missing. Ensure each table has a descriptive legend.
Discussion
The discussion provides insightful interpretation and situates findings within the broader literature. However, some additional considerations may strengthen it:
The “pill of knowledge” approach seems promising, but how might its effects persist over time? Have similar studies reported long-term knowledge retention or behavior change?
Beyond increased knowledge and vaccination intention shortly after the intervention, what evidence exists about actual behavior change? Is there regional data showing post-intervention changes in vaccination rates?
These points could be addressed to contextualize potential long-term impact and reinforce the study’s implications for public health practice.
Author Response
Please see the attached cover letter for a point-by-point review of your comments. Thanks

Reviewer 2 Report
Comments and Suggestions for Authors
The topic is very timely and definitely relevant to public health, especially now with the focus on improving HPV vaccination rates across Europe. The authors worked with a large and quite diverse sample, and the intervention is simple, practical, and low-cost, which makes the study potentially useful for policy or everyday practice.
1. Study design and methodology
The basic design is appropriate for this kind of educational intervention, but there are a few methodological points that really need to be explained more clearly in the manuscript:
Lack of a control group: Because this is only a pre/post design, it’s hard to separate the actual effect of the intervention from test–retest bias or other influences. The authors do mention some limitations, but the impact on internal validity should be discussed in a more direct way.
Questionnaire validation: The manuscript mentions a Delphi process and gives Cronbach’s alpha, but the questionnaire shown in the Supplementary Material looks longer than the “17 items” described in the text. It’s not very clear what was actually used. The authors should explain which items were included in the scoring and give a bit more detail about the validation process (especially), item selection, and reliability.
Potential biases/limitations (maybe it would be good to address these errors):
– Parental consent may have introduced selection bias, favoring families already positive toward vaccination;
– Giving both questionnaires so close together increases the chance of memory effects;
– Having healthcare professionals deliver the intervention might influence students’ intention to vaccinate simply because of the authority figure.
2. Statistical analysis
There is a lot of statistical analysis in the paper, probably too much. The amount of subgroup comparisons and p-values makes it hard to see what the most important results actually are.
The many subgroup analyses and repeated p-values make the Results section confusing and quite difficult to follow;
The tables are very dense and not easy to read. Some of them should probably be moved to the Supplementary Material;
Check for multicollinearity.
3. Results
The Results section is much longer than the Introduction or the Discussion, and it’s overloaded with numbers, repeated comparisons, and complicated tables. As it is now, it’s really difficult to understand the main take-home messages: the improvement in knowledge, the changes in intention to vaccinate, and the main predictors.
I strongly suggest rewriting this entire section, focusing on:
- the basic descriptive findings,
- the main pre/post differences,
- the important findings from the regression models.
4. Discussion
The Discussion feels unfocused and too long. There are repeated paragraphs, and a lot of background information that should actually be in the Introduction. In some places, the authors repeat numerical results already described earlier.
A clearer and more standard structure would really help. For example:
1. Short restatement of the main findings
2. Interpretation of what they mean
3. Comparison with previous studies (but more focused)
4. Strengths
5. Limitations
6. Implications for practice and future research
5. Language
The English needs some editing. There are several awkward sentences and places where the meaning isn’t completely clear (maybe the amount the information in Results makes it even harder to read). A careful language check would improve the manuscript a lot.
The English writing also needs improvement. There are several unclear sentences throughout the manuscript, and an important language edit would make the paper much easier to read.
Author Response

(The authors gave the same response as above.)

Reviewer 3 Report
Comments and Suggestions for Authors
Thank you for the opportunity to review the work ID: vaccines-4002438.
Major comments (listed in a logical order, based on the structure of the work):
Line 59: Add detailed information on the implementation of HPV vaccination in the observed population, including whether it was mandatory or not during the entire observed period, whether it was free or paid for, whether the delivery strategy of HPV vaccines was school-based or facility-based or mixed, etc. Cite appropriate references.
Line 359: Define `Study population`.
Line 359: Define type `Study sample`.
Line 359: Define `Participation rate`. Specify whether the population of participants and non-participants differed significantly by gender and age. Also, indicate whether all schools were equally represented in the Study sample.
Line 359: Specify `Study variables`.
Line 359: Specify `Study outcomes`.
Line 359: Define, I quote the title of Table 1 (on Lines: 98-99) `‡ HPV awareness is measured on a scale from 0 to 5, with higher scores indicating greater awareness.`. What items are included in determining this score?
Line 359: Questionnaire is not in English. Attach the English version.
Lines 101-146: Very important information is described in this text, so it must be presented as Tables/Figures.
But, during the review of the work itself, it is noted that this work is not complete.
Note: Since Supplementary materials and Supplementary Figures are mentioned in this manuscript, and are not fully available in this paper, it is not possible to fully review these for the paper as a whole.
Correct this, so that all relevant parts of this paper are available for review and decision-making.
Author Response

(The authors gave the same response as above.)

Round 2
Reviewer 3 Report
Comments and Suggestions for Authors
Thank you for the opportunity to re-review the work ID: vaccines-4002438.
The authors took my comments into account and made corrections in the revised version of this paper.
I believe that this paper is now much clearer, readable and informative.
I thank the authors for the effort they put into the revision of this work.